# Reference Values for Sagittal Clinical Posture Assessment in People Aged 10 to 69 Years

**DOI:** 10.3390/ijerph20054131

**Published:** 2023-02-25

**Authors:** Oliver Ludwig, Carlo Dindorf, Jens Kelm, Steven Simon, Felix Nimmrichter, Michael Fröhlich

**Affiliations:** 1Department of Sport Science, Rheinland-Pfälzische Technische Universität Kaiserslautern-Landau (RPTU), 67663 Kaiserslautern, Germany; 2Chirurgisch-Orthopädisches Zentrum, 66557 Illingen, Germany; 3Institute of Sport Science, Universität des Saarlandes, 66041 Saarbrücken, Germany

**Keywords:** posture, normative data, *flèche cervicale*, *flèche lombaire*, lordosis, kyphosis

## Abstract

Poor posture is a well-known problem in all age groups and can lead to back pain, which in turn can result in high socio-economic costs. Regular assessment of posture can therefore help to identify postural deficits at an early stage in order to take preventive measures and can therefore be an important tool for promoting public health. We measured the posture of 1127 symptom-free subjects aged 10 to 69 years using stereophotogrammetry and determined the sagittal posture parameters *flèche cervicale* (FC), *flèche lombaire* (FL), and kyphosis index (KI) as well as the values standardized to the trunk height (FC%, FL%, KI%). FC, FC%, KI, and KI% showed an increase with age in men but not in women, and a difference between the sexes. FL remained largely constant with age, although FL% had significantly greater values in women than men. Postural parameters correlated only moderately or weakly with body mass index. Reference values were determined for different age groups and for both sexes. Since the parameters analyzed can also be determined by simple and non-instrumental methods in medical office, they are suitable for performing preventive checks in daily medical or therapeutic practice.

## 1. Introduction

Postural deficits are known to be common in children and adolescents as well as in adults in industrialized countries. Various studies show a prevalence of occurrence of postural deficits in adolescents ranging from 25% to 60% [1,2,3]. In their study, Maulik et al. found musculoskeletal deficits in 73.3% of adults [4]. The causes include prolonged and unfavorable sitting postures, which, in the long term, can lead to muscular weaknesses and imbalances [5]. Muscular imbalance, in turn, causes various changes in posture that manifest as postural weakness [6,7]. This can include a forward-shifted (protracted) head, which is usually associated with increased lordosis of the cervical spine [8,9] and increased kyphosis of the thoracic spine, which can lead to a hunchback [10] and affect breathing [11,12]. In the lumbar spine, an increased forward-tilted pelvis and a hollow back can develop [13]. Such poor posture places a high load on the facet joints of the spine, which can lead to increased joint degeneration, muscular tightness, and, ultimately, pain [14,15,16]. Back pain, on the other hand, is responsible for a large proportion of work absence days in Germany [17] and other countries [18,19,20], also affects children and adolescents [21,22,23], and is, meanwhile, considered a global health problem [24]. Thus, it is clear that postural imbalance is becoming a significant factor in maintaining public health.

In a comprehensive review, Potthoff and colleagues called for further study of sagittal postural alignment as one of several risk factors for the development of low back pain in adolescence [25]. Sugai and colleagues [26] furthermore demonstrated that measuring thoracic kyphosis in the elderly could predict a future decline in their activities of daily living. This shows that an analysis of sagittal posture is useful in all age groups. In order to detect postural deficiencies at an early stage and to take necessary preventive measures, regular assessment of posture is useful. This makes posture measurement an important tool for promoting public health. However, a major problem is determining when a sagittal spinal curvature can still be described as normal and when it should be considered a postural weakness requiring treatment [27]. In daily medical practice, an initial assessment is made only by visual inspection and is based on the examiner’s experience. Invasive procedures such as X-rays are only performed when pathology is suspected, due to radiation exposure, especially in children and adolescents. However, assessing the back contour by visual inspection is subjective and prone to error. For example, a comparative study in which 28 chiropractors, physical therapists, rheumatologists, and orthopedic surgeons were asked to assess the posture of subjects from lateral photographs reported only mediocre intrarater reliability (kappa = 0.50) and weak interrater reliability (kappa = 0.16) [28].

Therefore, in order to objectify the analysis and quantify the shape of the back and spine in the sagittal plane, various measurements have been defined that can be determined non-invasively, such as the kyphosis angle, according to Debrunner [29]. The depth of cervical and lumbar lordosis can be assessed relatively easily with the parameters *flèche cervicale* (French for ‘cervical arrow’) and *flèche lombaire* (‘lumbar arrow’), usually using a plumb bob and a ruler [30,31]. Since these measurements can be collected quickly and inexpensively and are also reliable and reproducible [32], they have high practical suitability [33].

However, the above-mentioned methods have the disadvantage that the person being tested must usually be touched lightly during the measurement, so it cannot be ruled out that he or she will reflexively change posture as a result of the touch. In the case of postural defects due to immobile anatomical changes in the vertebrae, such as in Scheuermann’s disease, it can be assumed that the contour of the back will not change much as a result of touch. However, this cannot be ruled out in the case of postural deficiencies, where the shape of the back is determined primarily by muscular (dys)balance. Therefore, measurement methods that work via mechanical scanning of the back (e.g., Debrunner’s kyphometer [34], SpinalMouse^®^ [35], ultrasound systems [36], manual inclinometers [37], flexicurve [38]) sometimes have limited informative value, especially in people sensitive to touch.

One approach to counter this fundamental methodological problem is to use a stereophotogrammetric system that can scan the back contour without contact and radiation exposure by means of light grids and digital cameras and record its spatial shape with a high degree of accuracy [37,39,40,41]. With this method, conclusions about the deformation of the underlying spine can be made based on the body surface using mathematical models. Fortin et al., in a comprehensive review, concluded that non-contact methods (such as digital photographs or 3D scans) provide an accurate and rapid way to perform clinical postural assessment [42].

Nevertheless, the problem remains of defining reference values on this basis in order to identify the types of deviations in postural weakness that do not yet represent serious pathology and therefore do not require the use of invasive measurement methods such as X-rays. Since postural parameters can develop over a lifetime and can differ between the sexes, it is helpful to have a practical database that provides reference values to support medical diagnoses. The studies of Ohlendorf et al., who defined reference values for individual target groups using stereophotogrammetry [43,44,45,46,47], should be mentioned here in particular. Furthermore, Wolf et al. [48] provided reference values for vertebral position in the thoracic and lumbar regions in women between 20 and 69 years of age using 3D stereophotogrammetry, and Huthwelker and colleagues collected reference data of three-dimensional spinal parameters in symptom-free subjects 18 to 70 years of age [49]. In a review, Ribeiro et al. summarized photogrammetrically determined reference data for posture in young women [50].

These recent works provide a suitable database for assessing posture in healthy individuals. However, stereophotogrammetry, in addition to its many measurement advantages, has the disadvantage of high investment cost and is therefore rarely used in many application areas. In contrast, photogrammetric posture analysis is based on digital photographs. Using this measurement method, Stolinski and colleagues, for example, established reference ranges of posture values for 7- to 10-year-old children [51], and Ludwig et al. provided reference values for a posture index for children and adolescents 6 to 17 years old [52]. Gong et al. used posture photographs to examine changes in various sagittal posture angles across an age range of 20 to 90 years, but with small sample sizes in each subgroup [53].

In summary, it can be stated that to the best of the authors’ knowledge, there is no study to date that covers all of the following criteria and is thus suitable for screening and preventive examinations of posture in everyday medical, physiotherapeutic, or occupational practice (compare also [31]):

(1) The parameters should be reliably and validly measurable with simple methods, without extensive effort and cost-intensive instruments;

(2) The parameters should be able to cover several levels of the trunk in the sagittal plane (cervical, thoracic, lumbar spine);

(3) Reference values should exist that define when postural parameters deviate from the norm depending on age and sex.

The aim of this study was, therefore, to assess the distribution of three sagittal posture parameters (*flèche cervicale*, *flèche lombaire*, kyphosis index) based on sex in a symptom-free population including different age groups and thus to provide diagnostically useful reference values. A non-contact and accurate measurement method should be used, from which parameters can be extracted that also can be measured with sufficient accuracy in daily medical or therapeutic practice using simpler methods, for example, based on evaluating posture photos.

## 2. Materials and Methods

Subjects were recruited from companies and associations. They were informed verbally and in writing about the study procedure and the applicable data protection regulations and provided their written informed consent. In the case of minors, additional consent was obtained from their legal guardians. The study was conducted according to the guidelines of the Declaration of Helsinki and was approved by the institutional ethics committees.

### 2.1. Subjects

This study included 1150 participants. Exclusion criteria were acute complaints, chronic diseases of the spine or musculoskeletal system, previous spinal surgery, leg length discrepancies greater than 5 mm, and vertigo.

### 2.2. Measurements

The examinations were carried out on site at the participating companies or associations in a separate room during health days. Participants’ body height was measured with a stadiometer (Seca Stadiometer 213, Seca, Hamburg, Germany), weight was determined with a scale, and body mass index (BMI) was calculated. A mobile scanner (Balance 4D, Paromed bodybalance GmbH & Co KG, Neubeuern, Germany) was used to measure posture. The Paromed scanner uses a Vialux scanning unit (Vialux GmbH, Chemnitz, Germany), whose accuracy and reliability have been demonstrated (coefficients of variations 0.3–2.5% [54]). The system works by projecting a moving light stripe pattern (LED light source, wavelength 460 ± 20 nm) onto the subject’s back and achieves a spatial resolution of <1 mm. The validity of stereophotogrammetric measurement methods for assessing trunk shape has been confirmed in previous studies [55]. The subjects stood without shoes in their habitual posture with free upper body (women in bra) at a distance of about 2.30 m from the device. For our measurements, several anatomical landmarks were marked with white tape dots (diameter 12 mm) by the examiner beforehand: the spinous process of the seventh cervical vertebra (C7); the apices of cervical, thoracic, and lumbar spine curves; the spinous process of the first sacral vertebra (S1); the posterior superior iliac spines (PSIS); and the apices of the scapulae. Each scan was performed 4 times and the values obtained were averaged.

The anatomical landmarks were automatically recognized by the system and manually checked and confirmed by the examiner. The measurement system calculated the horizontal distances of the 3 vertices to the perpendicular through S1 (Figure 1). From this, the analyzed posture parameters were calculated as follows:*Flèche cervicale* FC = a + b;*Flèche lombaire* FL = b + c;Kyphosis index KI = (FC + FL)/2.
where a, b, and c represent the absolute values of the horizontal distances of the vertices from the perpendicular through S1 (Figure 1).

FC, FL, and KI were additionally calculated as percentages of trunk height (perpendicular distance between C7 and S1), denoted as FC%, FL%, and KI%.

### 2.3. Statistics

Subjects were divided into 10-year age groups. Exceptions were the two younger age groups: boys were grouped into 12–16 and 17–19 years and girls were grouped into 10–15 and 16–19 years. This unequal division was based on the different growth curves between sexes, since the growth spurt in boys normally does not begin until the age of 12, while in girls it begins at the age of 10, but also ends earlier [56]. Since only 3 individuals were older than 70 years, their data were excluded from further analysis; data on boys under 12 years (n = 11) and girls under 10 years (n = 4) were also excluded. Furthermore, separately for each sex-specific age group, extreme outliers exceeding 3 times the interquartile range were excluded (n = 5). Ultimately, data on 1127 subjects were used for final analysis.

Pearson correlation was used to calculate possible correlations between posture parameters. Because it was unclear whether the studied posture parameters depended on age and sex, a multivariate analysis of variance (MANOVA) was performed to test whether it was necessary to distinguish between age groups and sex. To avoid multicollinearity of the dependent variables, the non-normalized and normalized posture parameters were separately analyzed by MANOVA twice. The normal distribution of the data was checked using the Kolmogorov–Smirnov test. Because MANOVA is robust to deviations from the normal distribution [57], no transformation or non-parametric test was applied to data that were not normally distributed. The assumptions for MANOVA were checked and could be confirmed. Wilk’s lambda values are presented for the results of the MANOVA omnibus test. Effect size is reported as Cohen’s f, where f = 0.10 indicates a weak effect, f = 0.25 indicates a moderate effect, and f = 0.40 indicates a strong effect [58]. Bonferroni correction was applied to the post hoc tests. Adjusted *p*-values were reported and compared with an alpha level of 0.05.

In addition to age and sex, BMI is also reported in the literature as an influencing factor for posture, but the results are inconsistent [43,59]. In the present study, however, BMI was not included as separate factor in the MANOVA model because this would have resulted in very small subgroups and the focus of the paper is on age- and sex-specific differences. However, in order to evaluate sex-specific correlations of posture parameters with BMI, Pearson correlations were calculated.

Due to partly non-normally distributed data, means and medians, as well as corresponding bootstrap confidence intervals (1000 samples), were calculated to present group-specific values [60].

Calculations and visualizations were performed in SPSS (version 28, SPSS Inc., Chicago, IL, USA).

## 3. Results

### 3.1. Correlations of Postural Parameters

Table 1 shows the correlations of the posture parameters. KI, FC, and FL correlate very strongly with the corresponding normalized parameters KI%, FC%, and FL%. There are no or weak correlations between FL/FL% and FC/FC%. KI/KI% correlates strongly with the other parameters due to mathematical dependence.

### 3.2. Analysis of the Variables KI, FC, and FL

MANOVA showed significant differences for age (F(12, 2224) = 5.73, *p* < 0.001, n_p_^2^ = 0.03, f = 0.18, Wilk’s Λ = 0.94) and sex (F(2, 1112) = 80.20, *p* < 0.001, n_p_^2^ = 0.13, f = 0.39, Wilk’s Λ = 0.87) for the combined dependent variables KI, FC, and FL, but no interaction for age group and sex (F(12, 2224) = 1.60, *p* = 0.084, Wilk’s Λ = 0.98).

Post hoc univariate ANOVA showed statistically significant differences for both sex (KI: F(1, 1113) = 74.56, *p* ≤ 0.001, n_p_^2^ = 0.06, f = 0.25; FC: F(1, 1113) = 156.02, *p* < 0.001, n_p_^2^ = 0.12, f = 0.37) and age (KI: F(6, 1113) = 8.67, *p* < 0.001, n_p_^2^ = 0.05, f = 0.23; FC: F(6, 1113) = 9.40, *p* < 0.001, n_p_^2^ = 0.05, f = 0.23) for the dependent variables KI and FC. For FL, only differences between age groups were found (F(6, 1113) = 2.76, *p* = 0.01, n_p_^2^ = 0.02, f = 0.14) but not for sex.

An interaction effect of age and sex was found for FC (F(6, 1113) = 2.80, *p* = 0.01, n_p_^2^ = 0.02, f = 0.14), suggesting an increase in sex differences with increasing age. No further interaction effects were found (*p* > 0.05). Figure 2a,c,e shows the differences, including the results of the post hoc tests. Respective subgroup characteristics are presented in Table 2.

### 3.3. Analysis of Variables Normalized to Trunk Height: KI%, FC%, and FL%

MANOVA showed significant differences for age (F(12, 2224) = 2.87, *p* < 0.001, n_p_^2^ = 0.02, f = 0.14, Wilk’s Λ = 0.97) and sex (F(2, 1112) = 42.49, *p* < 0.001, n_p_^2^ = 0.07, f = 0.27, Wilk’s Λ = 0.93) for the combined dependent variables KI%, FC%, and FL%. An interaction effect of age and sex for the combined dependent variables could also be found (F(12, 2224) = 1.76, *p* = 0.05, n_p_^2^ = 0.01, f = 0.10, Wilk’s Λ = 0.98).

Post hoc univariate ANOVAs showed statistically significant differences for both sex (KI%: F(1, 1113) = 7.95, *p* = 0.005, n_p_^2^ = 0.01, f = 0.10; FC%: F(1, 1113) = 59.57, *p* < 0.001, n_p_^2^ = 0.05, f = 0.23) and age (KI%: F(6, 1113) = 2.89, *p* = 0.008, n_p_^2^ = 0.02, f = 0.14; FC%: F(6, 1113) = 5.19, *p* < 0.001, n_p_^2^ = 0.03, f = 0.18) for the dependent variables KI% and FC%. For FL%, statistically significant differences were found only between the sexes (F(1, 1113) = 18.14, *p* < 0.001, n_p_^2^ = 0.02, f = 0.14). We found interaction effects of age and sex for KI% (F(6, 1113) = 2.25, *p* = 0.036, n_p_^2^ = 0.01, f = 0.10) and FC% (F(6, 1113) = 2.74, *p* = 0.012, n_p_^2^ = 0.02, f = 0.14). Sex differences seem to increase with increasing age for both variables. Figure 2b,d,f illustrates the differences found, including the post hoc tests. The statistical characteristics of subgroups are shown in Table 3.

### 3.4. Correlations of the Posture Parameters with BMI

When considering sex, BMI is significantly correlated with the posture parameters FC, FC%, KI, and KI%, but following Cohen [58], only a moderately strong positive correlation could be identified for FC in both sexes (male: r = 0.35, *p* < 0.001 and female: r = 0.31, *p* < 0.001). Other correlations could be classified as weak (CI: male: r = 0.27, *p* < 0.001 and female: r = 0.20, *p* < 0.001; CI%: male: r = 0.15, *p* < 0.001 and female: r = 0.11, *p* = 0.02; FC%: male: r = 0.25, *p* < 0.001 and female: r = 0.24, *p* < 0.001; FL%: female: r = −0.11, *p* = 0.02). No correlations were found for FL (female: r = −0.04, *p* = 0.40 and male: r = 0.3, *p* = 0.49) and FL% for men (r = −0.05, *p* = 0.19). The anthropometric data of the subjects are shown in Table 4.

## 4. Discussion

The aim of the present study was to find reference values for postural parameters in the sagittal plane and to show possible differences with respect to age and sex.

It is well known that, especially at the transition points between lordosis and kyphosis, a particularly strong structural stress of the vertebrae and their joints, especially the facet joints, occurs [61,62]. Against this background, the postural parameters *flèche cervicale* and *flèche lombaire* are particularly interesting because they are calculated from the magnitude of differences in the maximum points of kyphosis and lordosis in the sagittal plane. Thus, according to the biomechanical analysis of vertebral loading, Bruno et al. emphasize that measuring kyphosis depth alone is not sufficient to provide information about the loading of vertebral segments [62]. Both *flèche* parameters were introduced by Stagnara in 1982 and are part of orthopedic diagnostics [33]. Their advantage is that they can be measured quickly and with sufficient accuracy in medical practice using very simple materials.

### 4.1. Flèche Cervicale

For *flèche cervicale* (FC), an increase over the life span in men was observed, with differences becoming statistically significant mainly in relation to the younger age groups (Figure 2). In women, differences were seen only with respect to the 30–39 age group. This trend remained even when normalized to trunk height (FC%). Here, differences between the older age groups (50–59 years) and the middle and younger age groups (<40 years) were particularly evident in men. Here a deterioration of posture was seen, possibly originating from increasing muscular insufficiency, especially of the dorsal cervical and thoracic muscle groups. However, since the studied subjects came from all occupational categories, it is not possible to assign them, for example, to more sedentary or standing activities. The difference between the sexes increased over the life span, as shown by the interaction effects.

Drzał-Grabiec et al. studied two age cohorts of women (20–25 and 60–90 years) and reported a statistically significant increase in thoracic kyphosis with age [63]. They cited osteoporotic changes in the spine, which are more highly prevalent in women and lead to increased thoracic spine kyphotic curvature, as an explanation. In the present study, an increase, although not significant, was found only in men, who had a significantly higher *flèche cervicale* than women. Gong et al. also examined changes in various sagittal postural angles photogrammetrically over an age range of 20 to 90 years [53]. They found a decrease in the neck angle they measured, corresponding to an increase in cervical lordosis with age. Similar findings have been confirmed in other studies [64,65,66].

Ohlendorf et al. described three-dimensional posture parameters in men and women in four age groups between 21 and 60 years in several publications [43,44,45,46,47]. With increasing age, thoracic kyphosis and lumbar lordosis angles (women > men) increased regardless of sex. Although the kyphosis angle is not directly comparable to *flèche cervicale*, our data also show an increase, but it is significant only between a few age cohorts. However, an age-related increase in thoracic kyphosis in women, as described by Ohlendorf et al., did not affect *flèche* values in the parameters we examined [43].

### 4.2. Flèche Lombaire

The *flèche lombaire* values, in turn, are stable over the course of adulthood. While the absolute values for *flèche lombaire* (FL) were found to be the same between men and women and no longer changed significantly with age, when normalized to trunk height (FL%), a significant difference was found between the sexes, with higher values in women. This result is in agreement with Huthwelker et al. [49], who also used stereophotogrammetry to study 201 subjects 18–70 years of age and found a higher lordosis angle in women than in men. In a review, Arshad et al. [67] demonstrated significantly greater lumbar lordosis in women than in men. Ohlendorf et al. also reported greater lordosis depth in women [43]. A greater degree of lumbar lordosis and thoracic kyphosis in women is considered to be, among other factors, a biomechanical adaptation process to breast size [68]. However, it is interesting to note here that for the *flèche lombaire* parameter we studied, a significant difference between the sexes was found only when normalized to trunk height. Since *flèche lombaire* is defined as the relative difference between the apices of the lumbar and thoracic spines and structural load on the spine depends more on the degree of curvature, especially in the thoracolumbar junction, than on absolute value [61,62], it could be a more meaningful parameter for preventive assessment than absolute lordosis and kyphosis depth.

### 4.3. Kyphosis Index

The kyphosis index, which was calculated mathematically from the *flèche* values, showed a marked increase with age, more pronounced in men than in women. This was mainly due to the increase in FC. The percentage kyphosis index (KI%) allows a distinction between the sexes, but it is largely stable with age. Thus, it allows a summary description of posture and should be investigated in further studies, especially in patients, to determine possible relationships between postural abnormalities and pain.

### 4.4. Body Mass Index

A comparison of the anthropometric data of the subjects who were examined in the present study with reference values from the German population [69,70] showed that the cohorts were representative of the overall population in terms of height, weight, and BMI. Correlations between individual posture parameters and BMI were only weak to moderate. Other studies also did not show homogeneous results; Ohlendorf et al. found an increase in curvature parameters with BMI, especially in women [43]. Kocur et al. also found only a moderate association between BMI and postural parameters of the head and neck region [59].

### 4.5. Limitations

The present study has several limitations. First, it must be emphasized that we should speak only of reference data and not normative data. Even though a total of more than 1000 healthy subjects were examined, an age dependency of posture parameters was found in some cases. Therefore, to provide true standard values, a larger sample would have to be collected for each subgroup. The size of the subgroups is also not homogeneous. In the younger age groups, there was significantly fewer female than male participants, so sex differences that are to be statistically validated should be viewed with caution. Furthermore, the studied subjects were symptom-free, so one cannot extrapolate the results to patients. It is known that back pain patients show changes in posture [16,71].

Even if the anthropometric data of the examined subjects in their respective age groups corresponded to the reference values of the German population, it is not possible to speak of a representative sample. For example, neither the level of physical activity nor the occupational activity was examined.

The measurement methodology used has been well studied in terms of its accuracy, and all test quality criteria have been confirmed [72,73]. Nevertheless, it cannot be excluded that errors occurred during the manual positioning of individual marker points. Attempts were made to avoid this by providing extensive training for the researchers involved, who all have many years of experience in the field of posture analysis and work according to a standardized procedure.

### 4.6. Application in Everyday Medical Routine

For the examination of posture in the sagittal plane, the X-ray method is certainly the most informative and, therefore, the gold standard. In terms of non-invasive measurement methods, 3D stereophotogrammetry has been established as a very accurate non-contact method [72]. Its disadvantage, however, is the associated high cost, which makes it inaccessible in daily medical or therapeutic practice. However, preventive examinations are of great importance in promoting public health [74]. For this reason, we deliberately chose simple postural parameters that also can be determined without elaborate instrumental measurement methods. The advantage of the posture parameters we examined in more detail is that they can be determined with suitable accuracy even with very simple tools (pendulum and ruler) quickly and without computational effort (Figure 3). This makes them useful for screening in different settings (general practice, pediatric or physiotherapy offices, clubs, schools, and companies in the context of occupational health), and they can make an important contribution in terms of preventing postural weaknesses.

Further studies should, in particular, delineate the reference values from pathological findings such as postural disorders. Likewise, an analysis of patients with different clinical symptoms would appear to be useful.

## 5. Conclusions

*Flèche cervicale* and *flèche lombaire,* as well as the kyphosis index calculated from it, represent easy-to-measure parameters with which posture can be quickly assessed. FL values appear to remain stable from adulthood onward, whereas FC, FC%, KI, and KI% increase with age in males, and sex differences become more pronounced with age. Taking into account age and sex differences, the posture can thus be assessed using the reference values presented in this study in order to initiate preventive or therapeutic measures if necessary.

## Figures and Tables

**Figure 1 ijerph-20-04131-f001:**
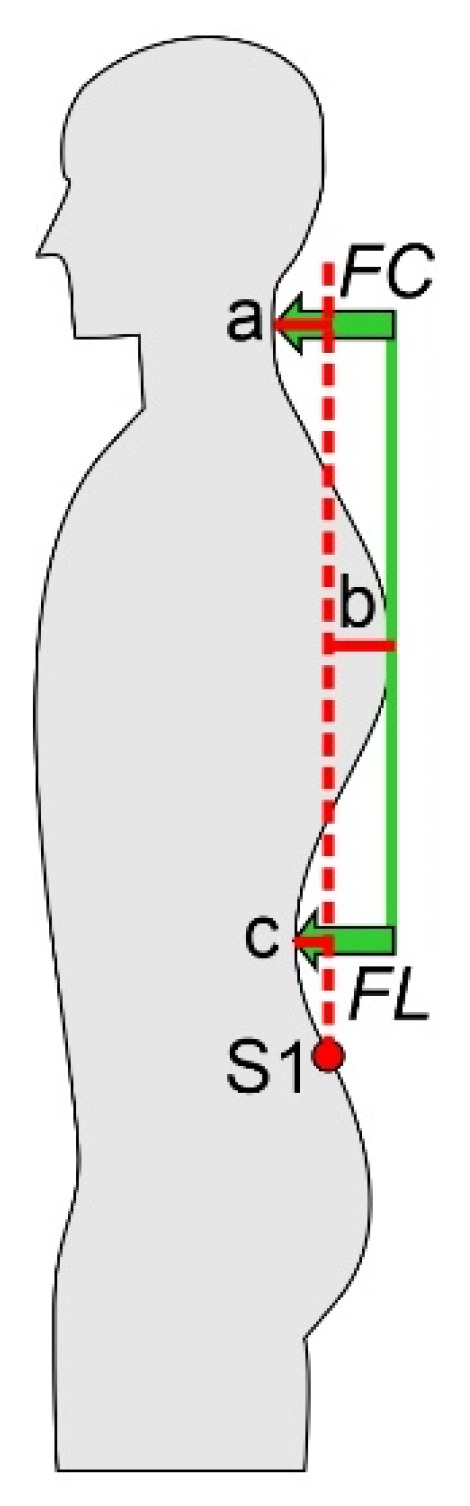
Red lines: horizontal distances of the 3 vertices to the perpendicular (dotted red line) through the first sacral vertebra (S1). Green arrows: *flèche cervicale* (FC) and *flèche lombaire* (FL).

**Figure 2 ijerph-20-04131-f002:**
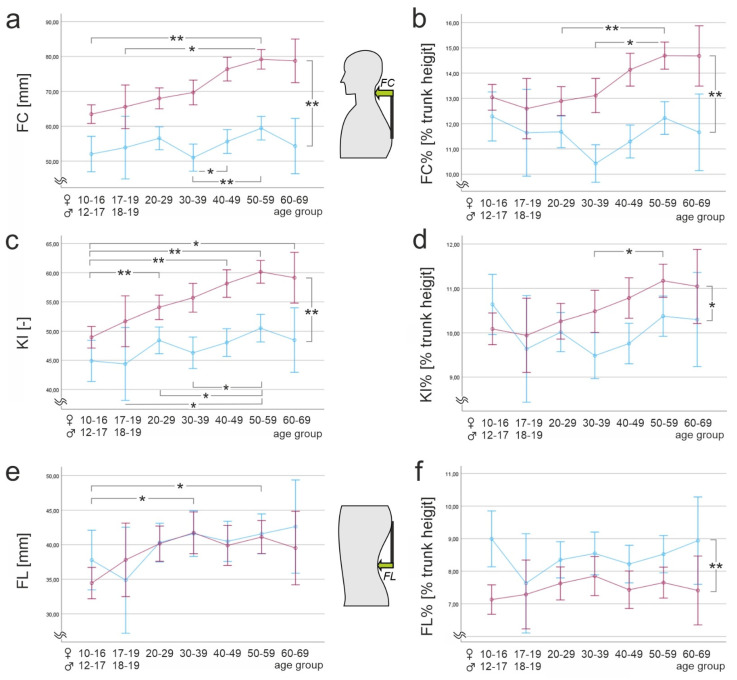
Posture parameters depending on age and sex. (**a**) *flèche cervicale* (FC); (**b**) *flèche cervicale* normalized to trunk height (FC%); (**c**) kyphosis index (KI); (**d**) kyphosis index normalized to trunk height (KI%); (**e**) *flèche lombaire* (FL); (**f**) *flèche lombaire* normalized to trunk height (FL%). Blue: women; red: men. Note that Y-scales do not start at 0 for better resolution. * *p* < 0.05, ** *p* < 0.001.

**Figure 3 ijerph-20-04131-f003:**
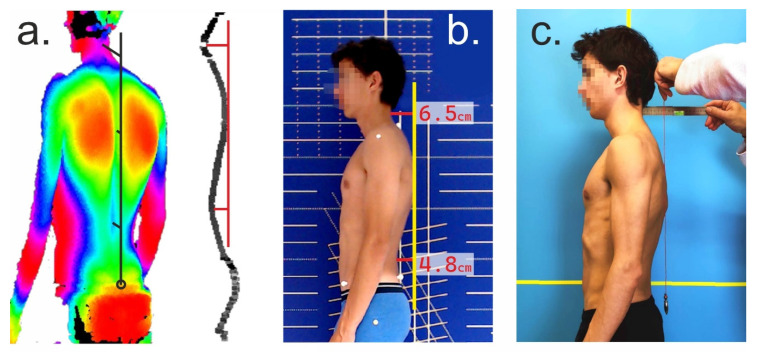
Different methods to determine the parameters *flèche cervicale* and *lombaire*. (**a**) 3D stereophotogrammetry, dorsal contour, and lateral profile; (**b**) 2D photogrammetric analysis (analysis of digital photos with graphics software); (**c**) plumb bob and ruler. Note that the ruler has a spirit level to check the horizontal position.

**Table 1 ijerph-20-04131-t001:** Correlation coefficients between postural parameters.

	KI	KI%	FC	FC%	FL
**KI%**	0.91 **				
**FC**	0.82 **	0.70 **			
**FC%**	0.75 **	0.78 **	0.94 **		
**FL**	0.66 **	0.66 **	0.11 **	0.078 **	
**FL%**	0.55 **	0.67 **	−0.01	0.05	0.96 **

** *p* < 0.001.

**Table 2 ijerph-20-04131-t002:** Reference data of non-normalized posture parameters. Non-normally distributed data according to the Kolmogorov–Smirnov test are shown in italics. FC, *flèche cervicale*; FL, *flèche lombaire*; KI, kyphosis index.

Parameter	Age Group (Years)	Sex	n	Mean	Median	25th Percentile	75th Percentile	Mean Lower 95% CI	Mean Upper 95% CI	Median Lower 95% CI	Median Upper 95% CI
KI (cm)	10–15	Female	44	44.90	44.50	37.11	50.70	41.54	48.68	40.45	48.20
	12–16	Male	159	48.96	49.80	40.35	57.05	47.07	50.82	47.35	51.10
	16–19	Female	14	44.38	42.85	39.45	50.59	40.61	48.15	40.15	49.80
	17–19	Male	29	51.69	49.80	45.28	58.58	48.34	55.29	46.90	55.35
	20–29	Female	105	48.43	47.50	42.48	54.35	46.43	50.45	45.80	49.40
		Male	126	54.07	52.85	44.79	63.16	52.06	56.47	51.95	55.40
	30–39	Female	75	46.29	46.30	40.15	53.60	43.90	48.53	44.20	48.70
		Male	90	55.70	56.88	47.10	63.95	53.01	58.24	52.35	58.80
	40–49	Female	97	48.05	46.70	40.20	56.30	45.45	50.63	44.15	49.92
		Male	98	58.14	59.05	49.61	66.86	56.29	60.14	54.65	61.84
	50–59	Female	99	50.51	49.30	40.90	59.10	47.86	52.96	46.20	51.55
		Male	144	60.15	59.20	50.80	68.48	58.04	62.19	55.90	61.35
	60–69	Female	18	48.46	47.70	37.83	55.75	41.82	55.18	41.27	55.22
		Male	29	59.14	56.95	47.53	67.93	54.56	64.14	53.81	64.23
FL (cm)	10–15	Female	44	37.78	38.00	29.53	45.45	33.91	41.77	32.40	41.55
	12–16	Male	159	34.45	34.60	25.30	44.10	32.38	36.34	31.00	36.80
	16–19	Female	14	34.88	34.15	29.30	40.63	31.70	38.36	29.30	40.30
	17–19	Male	29	37.81	36.90	28.15	50.40	31.91	43.34	30.30	47.40
	20–29	Female	105	40.31	40.40	30.90	48.45	37.62	42.92	37.50	42.10
		Male	126	*40.15*	*39.85*	*30.15*	*48.38*	*37.66*	*42.76*	*36.50*	*42.00*
	30–39	Female	75	41.60	43.20	32.50	49.00	38.29	45.13	37.40	45.35
		Male	90	41.72	39.25	30.30	54.10	38.66	45.06	36.10	44.05
	40–49	Female	97	40.49	39.60	31.00	51.65	37.74	43.43	36.20	42.90
		Male	98	39.89	41.00	29.48	50.58	37.11	42.59	35.30	43.75
	50–59	Female	99	41.57	38.50	31.40	51.00	38.52	44.57	37.20	44.39
		Male	144	*41.11*	*40.40*	*30.60*	*52.18*	*38.86*	*43.49*	*38.85*	*43.00*
	60–69	Female	18	42.62	44.60	27.38	53.90	33.64	50.85	28.00	51.80
		Male	29	39.52	42.60	30.30	47.20	34.83	44.10	32.15	45.10
FC (cm)	10–15	Female	44	52.02	52.80	39.83	63.95	47.64	56.63	45.80	56.10
	12–16	Male	159	63.46	64.10	53.60	72.50	61.09	65.74	60.05	67.30
	16–19	Female	14	*53.88*	*45.10*	*41.25*	*67.45*	*46.40*	*62.24*	*41.30*	*65.40*
	17–19	Male	29	65.57	65.20	56.40	72.50	60.12	71.03	58.90	68.75
	20–29	Female	105	56.54	55.30	45.95	65.85	53.71	59.30	51.61	60.25
		Male	126	67.99	65.70	55.78	78.65	64.94	71.28	62.60	70.05
	30–39	Female	75	50.98	50.30	41.20	60.00	46.93	55.08	47.10	53.80
		Male	90	69.68	69.10	58.78	80.85	66.42	72.96	64.50	72.70
	40–49	Female	97	55.61	55.60	43.00	66.00	52.00	59.01	50.50	59.30
		Male	98	69.68	75.65	66.13	90.13	73.15	79.49	71.00	81.90
	50–59	Female	99	59.44	58.80	47.60	70.90	56.01	63.00	56.60	63.49
		Male	144	79.19	75.90	65.98	91.40	75.95	82.37	73.15	81.24
	60–69	Female	18	54.31	54.70	40.68	67.13	42.82	65.21	41.55	63.76
		Male	29	78.76	78.30	67.35	94.90	71.72	85.97	69.40	91.40

**Table 3 ijerph-20-04131-t003:** Reference data of posture parameters normalized to the trunk height. Non-normally distributed data according to the Kolmogorov–Smirnov test are shown in italics.

Parameter	Age Group (Years)	Sex	n	Mean	Median	25th Percentile	75th Percentile	Mean Lower 95% CI	Mean Upper 95% CI	Median Lower 95% CI	Median Upper 95% CI
KI% (% trunk height)	10–15	Female	44	10.64	10.27	8.56	12.27	9.85	11.48	9.39	11.15
	12–16	Male	159	10.09	10.35	8.60	11.39	9.72	10.45	9.87	10.62
	16–19	Female	14	*9.64*	*10.30*	*8.32*	*10.93*	*8.85*	*10.39*	*8.34*	*10.92*
	17–19	Male	29	9.94	9.66	8.59	11.47	9.27	10.58	8.74	11.21
	20–29	Female	105	10.01	9.97	8.70	11.11	9.60	10.44	9.39	10.47
		Male	126	*10.26*	*9.98*	*8.46*	*11.90*	*9.83*	*10.77*	*9.65*	*10.47*
	30–39	Female	75	9.48	9.49	8.57	10.73	9.05	9.92	9.14	9.77
		Male	90	10.48	10.43	9.10	11.59	10.05	10.92	9.86	11.01
	40–49	Female	97	9.76	9.65	8.27	11.30	9.26	10.26	9.24	10.22
		Male	98	10.78	10.99	9.62	12.12	10.41	11.15	10.41	11.33
	50–59	Female	99	10.37	10.24	8.52	12.03	9.81	10.88	9.50	10.90
		Male	144	*11.17*	*10.93*	*9.64*	*12.57*	*10.80*	*11.54*	*10.37*	*11.49*
	60–69	Female	18	10.30	10.30	8.54	12.25	8.85	11.68	9.32	11.95
		Male	29	*11.04*	*10.38*	*8.82*	*12.29*	*10.17*	*11.92*	*9.55*	*12.10*
FL% (% trunk height)	10–15	Female	44	8.99	8.45	7.13	11.27	8.14	9.86	7.63	9.66
	12–16	Male	159	7.13	6.97	5.06	8.74	6.70	7.60	6.42	7.48
	16–19 17–19	Female	14	7.63	7.44	6.45	8.37	6.74	8.62	6.45	8.24
		Male	29	7.29	6.99	5.41	9.50	6.21	8.44	5.54	9.20
	20–29	Female	105	8.35	8.44	6.51	10.47	7.81	8.92	7.52	8.98
		Male	126	*7.62*	*7.33*	*5.79*	*9.31*	*7.11*	*8.14*	*7.08*	*8.25*
	30–39	Female	75	8.54	8.67	6.59	10.49	7.83	9.22	7.59	9.33
		Male	90	7.85	7.50	5.72	9.98	7.24	8.49	6.85	8.12
	40–49	Female	97	8.22	8.04	6.13	10.39	7.62	8.81	7.53	8.80
		Male	98	7.43	7.49	5.37	9.44	6.84	7.97	6.45	8.31
	50–59	Female	99	8.52	8.06	6.69	10.68	7.94	9.12	7.70	9.04
		Male	144	*7.65*	*7.85*	*5.78*	*9.59*	*7.22*	*8.11*	*7.07*	*8.12*
	60–69	Female	18	8.94	9.64	5.54	11.34	7.41	10.41	5.73	11.07
		Male	29	7.41	7.37	5.74	9.31	6.51	8.28	5.83	8.60
FC% (% trunk height)	10–15	Female	44	12.29	11.98	9.77	14.12	11.32	13.31	10.78	13.78
	12–16	Male	159	*13.04*	*13.22*	*11.13*	*15.13*	*12.57*	*13.53*	*12.70*	*13.72*
	16–19	Female	14	11.64	11.61	8.62	13.97	10.22	13.21	8.69	13.89
	17–19	Male	29	12.60	12.74	10.76	13.44	11.65	13.51	11.72	13.26
	20–29	Female	105	11.68	11.70	9.53	13.61	11.11	12.24	10.87	12.25
		Male	126	*12.89*	*12.37*	*10.59*	*14.86*	*12.33*	*13.43*	*11.77*	*13.04*
	30–39	Female	75	10.42	10.28	8.35	12.44	9.63	11.22	9.65	10.93
		Male	90	13.11	12.82	11.42	15.08	12.54	13.67	12.17	13.68
	40–49	Female	97	11.30	11.06	9.04	13.73	10.59	12.03	10.54	12.09
		Male	98	14.13	13.90	12.45	16.21	13.54	14.71	13.24	14.57
	50–59	Female	99	12.22	12.17	10.06	14.67	11.48	12.97	11.60	12.83
		Male	144	14.69	13.94	12.37	16.65	14.13	15.25	13.61	15.19
	60–69	Female	18	11.66	10.85	8.55	14.47	9.17	14.27	8.71	14.35
		Male	29	14.68	14.75	12.44	16.68	13.40	15.94	13.09	15.85

**Table 4 ijerph-20-04131-t004:** Anthropometric data of subjects. BMI, body mass index.

		Female		Male	
	Age Group	M	SD	M	SD
**Height (cm)**	12–16 ♂ 10–15 ♀	159.83	9.55	173.23	8.63
	17–19 ♂ 16–19 ♀	170.29	7.00	179.33	7.42
	20–29	167.53	6.59	180.42	6.08
	30–39	166.80	7.46	180.17	7.22
	40–49	166.89	6.33	180.42	7.37
	50–59	166.38	6.10	178.83	6.57
	60–69	162.33	5.36	178.48	7.49
**Weight (kg)**	10–15	49.11	10.93	61.03	11.48
	16–19	62.10	8.17	71.63	12.88
	20–29	65.54	12.26	81.64	11.55
	30–39	71.96	18.09	84.90	12.59
	40–49	69.36	11.87	89.43	17.31
	50–59	69.86	14.71	88.19	13.94
	60–69	68.17	15.16	84.83	15.07
**BMI (kg/m²)**	10–15	18.99	2.64	20.21	2.78
	16–19	21.39	2.08	22.13	2.88
	20–29	23.34	4.20	25.08	3.38
	30–39	25.79	5.82	26.15	3.70
	40–49	24.90	4.02	27.41	4.84
	50–59	25.24	5.16	27.55	3.86
	60–69	25.92	6.17	26.62	4.40

## Data Availability

The data will be made available upon justified request.

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
