# Peer review of "Reference Values for Sagittal Clinical Posture Assessment in People Aged 10 to 69 Years"

_ijerph, 2023, doi:10.3390/ijerph20054131_

Round 1
Reviewer 1 Report
Since the articel is in English, those posture parameters must be in English as well unless there is no English words for them.
To use the data for reference of measuring posture, Certain numbers should be recommended for certain age group to be used as reference. However, it is missing in conclusion
The measurement of data is good to be applied for a posture.
good number of N to present reference for healthy population.
However, No comparison between healthy and not make the result questionable to be applied clinically.

Reviewer 2 Report
The study presents the results of a study on posture assessment using stereophotogrammetry. The study included 1127 subjects aged 10 to 69 years and measured the sagittal posture parameters flèche cervicale (FC), flèche lombaire (FL), and kyphosis index (KI). The results showed that the values of FC, FC%, KI, and KI% increased with age in men but not in women and that there was a difference between the sexes. The correlation between the postural parameters and body mass index was moderate to weak. The study also determined reference values for different age groups and both sexes.
The authors highlight the importance of regular posture assessment as a tool for promoting public health and preventing back pain. The use of stereophotogrammetry is a reliable method for posture assessment, but the results can also be determined by simple non-instrumental methods in medical office.
Overall, the study provides valuable information on the changes in posture parameters with age and the difference between the sexes, and highlights the importance of regular posture assessment in daily medical or therapeutic practice. Further research is needed to validate the results and to determine the long-term impact of posture on public health and back pain.
Author Response
No suggestions, thank you very much.
Reviewer 3 Report
The manuscript entitled "Reference values for sagittal clinical posture assessment in people aged 10 to 69 years" assess the distribution of three sagittal posture parameters to provide useful reference values.
I present my comments and suggestions for changes in relation to the following parts of the article.
Overall, the same terms( fleche cervicale and fleche lombaire) are used differently in the abstract, keywords, methods, results, and discussion. If the terms mean the same thing, use the same expression.
Abstract
- The same sentences used in the introduction are in the abstract (Regular assessment of posture ...). I think it is better to avoid using the same expression as it is.
Introduction
- Unlike Abstract and Keywords, fleche cervicale and fleche lombaire are italicized. Is there a reason it's italicized?
Materials and Methods
- (They were informed verbally and ... their written informed consent.) These sentences seems to have grammatical problems. I think you need to correct the part where there is a problem with the grammar.
- It was mentioned that sagittal clinical posture measurements were performed on people aged 10 to 69 years, in the title and abstract. However, data on boys under 12 years (n = 11) were excluded. Also, the exact number of people in each group was not indicated. It would be better if you mention the number of people in each group.
- Please add explanations for a, b, c, d, e, f in Figure 2.
Conclusions
- I think it is better to present a conclusion about this manuscript based on the results.
Reviewer 4 Report
General comments
First of all, I am grateful for the opportunity to review this manuscript. All comments made below are intended to improve the scientific quality of your manuscript. The research carried out aims to establish normal values for three assessments of the spine in the sagittal plane in the population from 10 to 69 years (men and women). The main result corresponds to the proposed aim. The large number of participants in the sample (N=1227) is especially noteworthy.
In my point of view, there are some claims that need to be supported with references. In addition, in some sections of the manuscript, the third person is not used and phrases such as "we observed, our study..." are used.
The most important aspect that I have doubts about is the validity and reliability of the instrument used to make the assessments. The methods are correctly described, but there is no information about the device used "Balance 4D" (reliability, validity, accuracy...).
Specific comments
Introduction
"...work absence days in Germany and other countries (17)" - please include references from other European or world countries (example:reference 21).
In the paragraph of the different valuation methods, include references with studies in SpinalMouse, Inclinometers, Kyphometers...).
Measurement - It is necessary to reference if there are studies that have used this device, as well as the levels of validity, reliability and accuracy.
Table 3. Male sometimes was written in italics.
4.1 Flèche cervicale - "we observerd...." - Please, write in the third person for a more scientific writing. "here we see..." (there are similar affirmations in discussion section.).
Limitations
"our study..." - the present study.... the sample size and representativeness with respect to the German population have not been described. The levels of physical activity are unknown, as well as the work/profession of the sample studied.
